# Safety Analysis of Bevacizumab in Ovarian Cancer Patients

**DOI:** 10.3390/jcm12052065

**Published:** 2023-03-06

**Authors:** Yingwen Wang, Hao Lin, Yuche Ou, Hungchun Fu, Chingchou Tsai, Chanchao Chang Chien, Chenhsuan Wu

**Affiliations:** 1Department of Obstetrics and Gynecology, Kaohsiung Chang Gung Memorial Hospital, Kaohsiung City 833, Taiwan; 2Department of Obstetrics and Gynecology, Chia-Yi Chang Gung Memorial Hospital, Chiayi County 613, Taiwan; 3Graduate Institute of Clinical Medical Sciences, Chang Gung University, Taoyuan City 333, Taiwan

**Keywords:** bevacizumab, ovarian cancer, safety, adverse event

## Abstract

Bevacizumab (BEV) is beneficial for ovarian cancer patients, but the real world’s patient settings differ from those in clinical trials. This study tries to illustrate adverse events in the Taiwanese population. Patients with epithelial ovarian cancer treated with BEV at Kaohsiung Chang Gung Memorial Hospital between 2009 and 2019 were retrospectively reviewed. The receiver operating characteristic curve was adopted to identify the cutoff dose and the presence of BEV-related toxicities. A total of 79 patients receiving BEV in neoadjuvant, frontline, or salvage settings were enrolled. The median follow-up time was 36.2 months. Twenty patients (25.3%) had “De novo” hypertension or the worsening of a preexisting one. Twelve patients (15.2%) had “De novo” proteinuria. Five patients (6.3%) had thromboembolic events/hemorrhage. Four patients (5.1%) had gastrointestinal perforation (GIP), and one patient (1.3%) had wound-healing complications. Patients with BEV-related GIP had at least two risk factors for developing GIP, most of which were conservatively managed. This study revealed a compatible but distinct safety profile from those reported in clinical trials. The presence of BEV-related changes in blood pressure showed a dose-dependent trend. Most of the BEV-related toxicities were managed individually. Patients with potential risks for developing BEV-related GIP should use BEV with caution.

## 1. Introduction

Ovarian cancer is the leading cause of gynecological cancer death and the fifth most prevalent female cancer death in developed countries [1,2]. Though ovarian cancer includes a heterogenous group of tissue origin, 90% of ovarian cancers are epithelial ovarian cancers (EOC). Sixty-five percent of EOC is diagnosed at an advanced stage [3]. Though there is a lesser proportion of advanced stage EOC in Taiwan (44.8%) [4], the overall 5-year survival rate for all stages only increases to 61.3%. The 5-year survival rate is about 40% globally [5,6]. The aggressive behavior of EOC results in its poor prognosis despite providing standard care, which includes optimal debulking and systemic chemotherapy [7,8]. During surveillance after the standard treatment for EOC, the recurrence rate is about 75%, and most relapses and cancer deaths are related to the development of drug resistance [9]. To indicate the severity of recurrence, patients with recurrent EOC are usually further classified by the platinum-free interval (PFI). The response rate to second-line chemotherapy may be over 60 % for patients with platinum-very-sensitive recurrence (PFI ≥ 12 months) but this drops to only 10 % for patients with platinum-resistant recurrence (PFI < 6 months) [10,11].

Owing to the poor prognosis of EOC, various target agents have been developed, such as anti-vascular endothelial growth factor (anti-VEGF) [12] and poly ADP-ribose polymerase inhibitor (PARP inhibitor) [13]. Bevacizumab (BEV) is a monoclonal neutralizing antibody for the VEGF-A ligand. The binding of BEV prevents the ligand–receptor interaction, which inhibits the angiogenesis pathway. Angiogenesis is crucial in tumor cell growth [12]. The GOG-218 study [14] and the ICON 7 trial [15] illustrated the efficacy of BEV maintenance therapy in newly diagnosed EOC with advanced stages. The OCEANS trial [16] and the GOG-213 study [17] showed the effectiveness of BEV in managing patients with platinum-sensitive recurrent EOC. The AURELIA trial indicates that adding BEV to chemotherapy significantly improved progression-free survival for patients with platinum-resistant recurrent EOC [18]. Based on the promising results of these clinical trials, the addition of BEV has become a well-established practice in the management of advanced and recurrent EOC [19].

Though these clinical trials demonstrated the efficacy and the safety of BEV, the ethnic setting and the histologic composition in the clinical trials were different compared to the condition in Asian countries [14,15,16,17,18]. The Asian population only accounted for 3.1–13.6% of the enrolled patients in the clinical trials, and most of the histologic types were serous carcinoma (69.0–85.3%) and endometrioid carcinoma (13.2–7.7%). However, the histologic composition is different in Asian countries. Taking clear cell carcinoma (CCC) as an example, it is usually viewed as a rare histologic type (2.9–8.3%) in Europe and North America, but accounts for approximately 25% of cases in Japan [20] and 18.5% in Taiwan [4]. In addition, many patients in real clinical settings usually have medical comorbidities, which could be excluded and not evaluated in the clinical trials. This study tried to illustrate the real-world incidence and the adverse events after exposure to BEV among patients with EOC in Taiwan and also illustrate potential strategies for managing BEV-related toxicities.

## 2. Materials and Methods

This retrospective cohort study was conducted at the Department of Obstetrics and Gynecology, Kaohsiung Chang Gung Memorial Hospital (KCGMH). Patients who had received BEV for cancer management from January 2011 to December 2019 were identified in our institutional database. The database, established in 2008, encompasses EOC, peritoneal cancer, and fallopian tube cancer. BEV can be used in combination with chemotherapy or used alone as a single agent. We enrolled patients that received BEV in the following scenarios: (1) as a front-line adjuvant chemotherapy after primary surgery, (2) as an upfront neoadjuvant chemotherapy followed by interval debulking surgery, and (3) as salvage chemotherapy. This study received approval from the Ethics Committee and the Institutional Review Board of KCGMH (IRB202101712B0).

The following demographic and clinical data were extracted from medical records: age at diagnosis, body mass index (BMI), comorbidity, cancer type, FIGO stage, histology, type of primary cancer management, type of primary surgery, chemotherapy regimen, response to platinum, the dose of BEV, toxicities, and the related information regarding the management of BEV-related toxicities. The Charlson Comorbidity Index (CCI) illustrates comorbidity. The disease was staged based on the 2014 FIGO staging system. Optimal debulking surgery was defined when the maximal diameter of the residual tumors was smaller than 1 cm; the others were defined as suboptimal debulking surgeries. Patients with recurrent disease were further categorized based on PFI. We analyzed the dose and duration of BEV administration. Patients receiving only one cycle of BEV were also enrolled in this investigation. The Common Terminology Criteria for Adverse Events (CTCAE) version 5.0 graded BEV-related toxicities, and all the BEV-related toxicities were stratified into six major categories based on the presentation of the enrolled patients and the current literature [21]. 

According to international guidelines and our institutional consensus, ovarian cancer patients were regularly followed up every 1–3 months for two years, every 3–6 months for another three years, and then every 6 months after five years. The office visit included a symptoms review and physical and pelvic examination. Cancer antigen 125 (CA-125) was checked at every visit, and a chest radiograph was arranged annually. Abdominal or pelvic computed tomography (CT) was arranged every 6–12 months during the initial two years and then arranged when clinically indicated.

Data were analyzed using SPSS version 22 (IBM, Armonk, NY, USA). Descriptive statistics were reported as the mean and standard deviation. The rate of significant adverse events was summarized. The receiver operating characteristic (ROC) curve was adopted to identify BEV’s cutoff dose and predict BEV-related toxicities.

## 3. Results

### 3.1. Clinico-Pathologic Characteristics of the Enrolled Patients

This study enrolled 79 patients (Table 1). The median follow-up time was 36.2 months. There were 72 patients (91.1%) having ovarian cancer, 4 patients (5.1%) having fallopian tube cancer, and 3 patients (3.8%) having peritoneal cancer. The median age at diagnosis was 56.1 years (range 19–85 years). There were 47 patients (59.5%) denying having any systemic disease before, and 32 patients (40.5%) had a mean CCI of 1.6. A total of 84.8% of the patients presented with advanced-stage conditions (stage III or IV). Among all the enrolled patients, the three major histologic types were high-grade serous carcinoma (59.2%), CCC (21.1%), and endometrioid carcinoma (5.3%). Forty-five patients (57%) received primary debulking surgery, while thirty (38%) received neoadjuvant chemotherapy followed by interval debulking surgery as the primary cancer treatment. Regarding surgery, 51 patients (64.6%) received optimal debulking surgery, and 24 (30.4%) received suboptimal debulking surgery. Among all the adjuvant chemotherapy regimens used in primary cancer treatment, platinum combined with paclitaxel (72.2%) was the most used. When analyzing patients with recurrent EOC and classifying them by platinum sensitivity, 65.6% were platinum-sensitive, and 32.8% were platinum-resistant.

### 3.2. Details of Patients Treated with BEV

Among the 79 patients, BEV was used in more than one clinical scenario (Table 2). Forty-eight patients (60.8%) received BEV in salvage treatment, 42 patients (53.2%) received BEV in frontline treatment, and 7 patients (8.9%) received BEV in neoadjuvant chemotherapy. A carboplatin–paclitaxel combination was the most common doublet used with BEV, while lipodox was the most common single agent combined with BEV. A total of 81.0% of the BEV dose was prescribed at 7.5 mg/kg with a 3-week interval. The mean accumulated dose of BEV was 4058 ± 3558 mg with a minimum of 200 mg and a maximum of 16,838 mg.

### 3.3. Prevalence and Distribution of the BEV-Related Adverse Events

The adverse events associated with BEV are summarized in Table 3. Among all the 46 adverse events, only eight (8.9%) were categorized as ≥grade 3 toxicities. When further analyzing the eight adverse events, three events that occurred in two patients (2.5%) might be categorized as grade 5 toxicities due to the association with death within 2 weeks. According to the results, the three most common adverse events were as follows: changes in blood pressure (20 patients, 25.3%), “De novo” proteinuria (12 patients, 15.2%), and thromboembolic events/hemorrhage (five patients, 6.3%). Regarding the different changes in blood pressure, 17 patients (21.5%) were newly diagnosed with hypertension, and three patients (3.8%) needed to modify the previous anti-hypertensive agents to control the elevated blood pressure after using BEV. All the patients who experienced “De novo” proteinuria were classified as grade 1–2 toxicity. Though the proportion of thromboembolism/hemorrhage and gastrointestinal perforation (GIP) was not common, these had a higher probability of leading to life-threatening outcomes.

### 3.4. No Significant Correlation between the Total BEV Dosage and the Adverse Events in Our Cohort

The ROC curve was adopted to identify the potential BEV cutoff dose and the presence of related toxicities. Aside from “change in blood pressure” and “De novo proteinuria”, all other adverse events were grouped for analysis. Figure 1a illustrates the correlation between “changes in blood pressure” and the total BEV dosage. The area under the curve (AUC) was 0.695, and the optimal cutoff value for predicting BEV-related changes in blood pressure was identified as 3958 gm with a sensitivity of 0.600 and a specificity of 0.655. When evaluating the “De novo proteinuria”, the AUC was 0.533, which was approximately 0.5, indicating no specific cutoff dose could show significant discrimination (Figure 1b). When investigating all other adverse events, including GIP, thromboembolic events/hemorrhage, wound-healing complications/fistula, and intra-abdominal infection (IAI), the AUC value was 0.476, suggesting that no cutoff dose could show its discrimination (Figure 1c).

### 3.5. Our Experience in Managing BEV-Related Toxicities

We summarized the clinical features of the patients experiencing adverse events after BEV exposure and the corresponding management. Aside from BEV-related hypertension, proteinuria, and other toxicities categorized in Table 3, the corresponding managements are illustrated in Table 4.

#### 3.5.1. Clinical Features of Patients Having BEV-Related Gastrointestinal Perforation

The condition of four patients that experienced BEV-related GIP was summarized (Table 4A), including the risk factors contributing to the development of GIP. Our patients suffering from BEV-related GIP all had at least two risk factors [22,23]. The dose was irrelevant to the development of GIP. According to the data, one patient experienced GIP after receiving only one cycle of BEV combined with neoadjuvant chemotherapy, while one patient experienced GIP after receiving 15 cycles of BEV combined with salvage chemotherapy. It was challenging to identify a definite perforation site among the four patients, and they were mainly managed with conservative treatment, which included Nulla per os (NPO), total parenteral nutrition (TPN), and broad-spectrum antibiotics. Only one patient received an exploratory laparotomy, but no specific perforation site could be found. The survival days after developing GIP varied, ranging from 6 days to 603 days.

#### 3.5.2. Clinical Features of Patients Having BEV-Related Thromboembolism (TE)

After excluding some patients who already had TE before using BEV, two patients were summarized in Table 4B. One had CCC, and the other had high-grade serous carcinoma. Their BMI was within normal ranges. Case 1 had TE at the right distal internal carotid artery, and she was managed with thrombectomy. Her survival was only nine days after the diagnosis of TE. She is also referred to in Case 4 of Table 4A, and the survival days between the two episodes were incoherent owing to the different timing of the diagnosis of GIP and TE. Case 2 had TE of the right frontal lobe, and she was managed by conservative treatment, which included close monitoring of vital signs, adequate intravenous fluid supplement, and proper blood pressure control.

#### 3.5.3. Clinical Features of Patients Having BEV-Related Bleeding

Three patients had BEV-related bleeding, and the clinical presentation varied among these patients. One presented as gum and nasal mucosal bleeding, which was managed by discontinuing BEV temporarily. One presented severe flank and back pain, and the CT scan illustrated a left renal subcapsular hematoma. This patient was also managed by discontinuing BEV. Pigtail drainage was once considered; after a complete discussion with the interventional radiologist and the urologist, the hematoma was left in situ for compression and to prevent further progression. The third patient presented with persistent abdominal pain and a drop in hemoglobin level, despite regular blood transfusions. With the help of an RBC nuclear scan, active bleeding originating from a metastatic liver tumor was identified, and the bleeding was finally controlled by trans-arterial embolization.

#### 3.5.4. Clinical Features of Patients Having Surgical Wound-Healing Complications or Fistula after the Exposure to BEV

One patient had a surgical wound infection with an intra-abdominal abscess that had accumulated beneath the wound. The interval between the last BEV dose and surgery was 50 days, and no bowel resection was performed during the debulking surgery. This patient was managed conservatively, which included intensive wound care, CT-guided drainage, and broad-spectrum intravenous antibiotics.

#### 3.5.5. Clinical Features of Patients Having IAI after the Exposure to BEV

Four patients had IAI after exposure to BEV. These patients’ clinical presentation was similar, including abdominal pain, fever, and leukocytosis. They were all successfully managed conservatively using broad-spectrum antibiotics, NPO, and TPN.

## 4. Discussion

In our analysis, there were patients using BEV on more than one occasion. Most of these patients used BEV in the frontline and salvage setting. According to the phase III randomized trial conducted by Pignata et al., adding BEV to salvage chemotherapy still showed benefits in progression-free survival for patients with platinum-sensitive recurrent ovarian cancer who had already received first-line platinum-based treatment, including bevacizumab [24]. In our study cohort, two patients extended BEV beyond the frontline and salvage setting and combined BEV in neoadjuvant chemotherapy. Though there is still uncertainty about adding BEV to neoadjuvant chemotherapy, the efficacy and safety of the combination have been illustrated in two phase II trials (GEICO 1205 and ANTHALYA) [25,26].

### 4.1. BEV-Related Hypertension

Hypertension is the most common adverse event associated with BEV. The incidence reported in clinical trials ranged from 22.9–41% of any grade (6–17.4% ≥ grade 3) [14,15,16,17]. In our study, 20 patients (25.3%) had de novo hypertension or worsening of a preexisting one after BEV exposure, and it seemed to be a dose-dependent trend for a BEV-related change in blood pressure (AUC: 0.695). If the accumulative BEV dose is above 3958 gm, there might be a BEV-related change in blood pressure with a sensitivity of 0.600 and a specificity of 0.655.

The possible mechanism of BEV-related hypertension has been illustrated [27]. BEV inhibits the VEGF signaling pathway and reduces the activity of nitric oxide, which is vital for vasodilatation; on the other hand, BEV enhances the vasomotor tone through the interaction with the endothelin system, and BEV promotes microvasculature transformation. The latter two factors lead to increased vascular resistance. Based on the above mechanism, it is not surprising that BEV-related hypertension is associated with a higher BEV dose. Based on the results of one meta-analysis, including 22 randomized control trials (RCT) which enrolled over 20,000 patients receiving chemotherapy ± BEV, it indicated that patients treated with high-dose BEV have a higher risk of developing hypertension [28]. Since the BOOST trial indicated no significant survival benefits for the prolonged use of BEV in advanced-stage ovarian cancer patients [29], physicians could keep the BEV use at a standard duration and minimize the risk of developing BEV-related hypertension.

BEV-related hypertension can develop at any time during the treatment, but it most commonly develops within half a year (4.6–6.0 months) after exposure to BEV [30]. The choice of antihypertensive agents should be individualized, and there is no clear recommendation for specific antihypertensive agents [27,30,31]. Some studies indicated that angiotensin-converting enzyme inhibitors (ACEi) might benefit hypertensive patients with proteinuria. Some studies suggested that calcium channel blockers might be effective in reducing vascular smooth muscle tone, which is impaired after the introduction of BEV [30,31].

### 4.2. BEV-Related Proteinuria

Proteinuria is also a common side effect following the use of BEV. The incidence of all-grade proteinuria is approximately 4.4–17.0% (0.5–8.5% ≥ grade 3) [14,15,16,17,18]. In this study, 15.2% of patients had de novo proteinuria or the worsening of a preexisting one after exposure to BEV. The presence of VEGF-A plays a vital role in maintaining the structure and function of the glomerular barrier. Decreased VEGF-A induced by BEV prevents the podocytes and endothelial cells from maturation and proliferation, consequently impairing the glomerular filtration barrier [32]. Though most patients have asymptomatic BEV-related proteinuria, severe proteinuria might contribute to renal damage and cardiovascular disease [32]. Antihypertensive agents such as ACEi or angiotensin receptor blocker (ARB) might be helpful in managing concomitant BEV-related hypertension and proteinuria [32,33].

### 4.3. BEV-Related Gastrointestinal Perforation and Intra-Abdominal Infection

Though the incidence of GIP is low, GIP is perhaps the most concerning adverse effect following BEV. According to the results of RCTs, it ranged from 0–1.8% of any grade [16,17]. The percentage of enrolled patients in the AURELIA trial and the GOG-218 study who were noted to have ≥ grade 2 GIP toxicity was 2.2–2.6% [14,18]. A total of 1.3% of patients were noted to have GIP toxicity ≥ grade 3 in the ICON7 study [15]. In our study, 3.8% of the patients had grade 1–2 GIP, and 1.3% had GIP toxicity ≥ grade 3. The exact pathogenesis of BEV-related GIP is still unclear. The proposed mechanism shares some similarities with the BEV-related arterial toxicities: (1) decreased flow to splanchnic microvasculature via thrombosis or vasoconstriction leading to GI ischemia and compromising healing after GI injury, and (2) necrosis of a tumor with the weakening of the intestinal wall [22].

In our study cohort, the patients suffering from GIP had at least two of the risk factors mentioned above. The risk factors for BEV-related GIP were underlying inflammatory bowel disease, small bowel resection at primary surgery, and large bowel resection at primary surgery [23]. Symptoms of bowel obstructions and bowel involvement by tumor were also identified as risk factors associated with GIP [22]. Owing to our study result, we will illustrate the risk of BEV-related GIP when counselling patients about BEV and also avoid using BEV in patients who have at least two risk factors or delay its use until clinically feasible.

Early identification and prompt management could limit the progression of peritonitis and sepsis. Physicians should be aware of the following presentation among ovarian cancer patients receiving BEV: abdominal pain, bowel obstruction, fever, and leukocytosis. The contrast-enhanced CT scan is a preferred diagnostic modality with its high sensitivity in detecting signs of perforation. Currently, there is no firm guideline on the management of BEV-related GIP. When a BEV-related GIP is detected early, it could be conservatively managed, including broad-spectrum antibiotics, abscess drainage, and bowel rest with NPO and intravenous nutrition. Nevertheless, treatment should be individualized, and surgical intervention should be considered clinically indicated [22,34].

In our study, four patients had IAI without a specified etiology. The IAI could potentially be a minor form of GIP since the adverse events could all be conservatively managed, and CT was not routinely arranged to evaluate the intra-abdominal condition.

### 4.4. BEV-related bleeding episode

The incidence of BEV-related bleeding was variable, ranging from 1.1% to 41.5% (5.7% ≥ grade 3) in RCTs [14,16,17,18], and the incidence was 3.8% in our study cohort. The increased risk of bleeding with BEV could also be explained by the inhibition of VEGF, which prevents the endothelial cells from regeneration, resulting in endothelial defects and presenting as hemorrhage in the end [28]. Two distinct types of bleeding have been described: mild spontaneous mucocutaneous bleeding and tumor-related severe bleeding. Our patients presented with renal subcapsular hematoma and metastatic liver tumor bleeding. The risk of bleeding depends on several factors, such as stages, thrombocytopenia, and comorbidities predisposing to bleeding. There is no established treatment for controlling bleeding episodes [28,35].

### 4.5. Other BEV-Related Vascular Adverse Events

In addition to hypertension, arterial adverse events include cerebral infarction, peripheral arterial thrombotic events, and myocardial infarction [28]. The incidence of arterial TE was 0.7–3.6% in RCTs [14,15,16,17,18]. In this study, two patients (2.5%) had cerebral infarction after BEV exposure. The potential mechanism is that BEV produces a prothrombotic status triggered by hypertension, resulting in endothelial injury and subsequent prothrombotic status [28].

BEV increased the risk of venous TE as well [28]. Patients with clear-cell-type EOC also have a 2.5-times more significant risk of disease-related TE [36]. It is known that there is a higher proportion of clear-cell-type EOC among the Asian population, and the ratio of CCC is 21.1% in our study. The higher composition of CCC might lead to an even higher risk of venous TE when these patients are managed with BEV. Thus, the relatively high risk of developing TE when providing BEV for patients with CCC should be fully discussed before using BEV.

### 4.6. BEV Dose and the BEV-Related Adverse Events

Current evidence indicates that BEV-related cardiovascular adverse events and proteinuria are dose-dependent [28]. Our study showed a dose-dependent trend for BEV-related changes in blood pressure (AUC: 0.695), but there is no significant correlation between the total BEV accumulative dose and the presence of other adverse events. One retrospective study found that higher cumulative doses of BEV are associated with cardiovascular-disease-related hospitalization (*p* = 0.048) [37]. One retrospective study conducted in the Northern Taiwanese population illustrated that the cumulative incidences of BEV-related hypertension would plateau at around 30% above the dose of 8080 mg. The BEV-related proteinuria cumulative incidence would plateau at about 35% above the quantity of 11,190 mg [38].

### 4.7. Limitation

Our study specifically focused on assessing the adverse events among patients with EOC in Taiwan. The results presented here are subject to some limitations. For example, our study was a retrospective, single-center design, including different clinical settings (upfront neoadjuvant, frontline, and salvage). In addition, only a limited number of cases was enrolled in this study. The incidence of adverse events might be underestimated since related laboratory tests and image exams are not prospectively and regularly followed. A prospective multi-centered study may be warranted to further illustrate the adverse events regarding BEV use among the Taiwanese population.

## 5. Conclusions

This retrospective study of the Taiwanese population revealed a compatible but distinct safety profile from those reported in clinical trials. The presence of BEV-related changes in blood pressure showed a dose-dependent trend, but the accumulative BEV dose did not show an obvious correlation with other adverse events. The pros and cons should be fully discussed before adding BEV to cancer treatment, and most of the BEV-related toxicities were managed individually. BEV should be used cautiously when patients potentially have risk factors for developing BEV-related GIP.

## Figures and Tables

**Figure 1 jcm-12-02065-f001:**
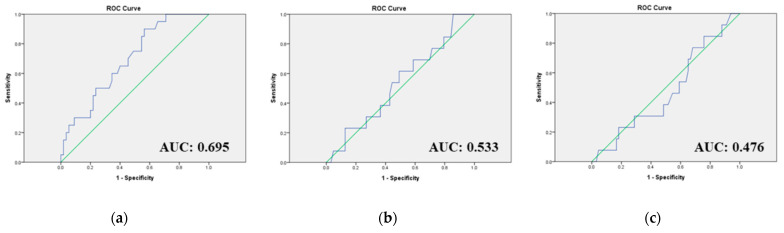
The receiver operating characteristic (ROC) curve analyzing the accumulative BEV dose and presence of BEV-related toxicities. (**a**) The accumulative BEV dose and the BEV-related changes in blood pressure. (**b**) The accumulative BEV dose and the BEV-related “De novo” proteinuria. (**c**) The accumulative BEV dose and other BEV-related toxicities (including GIP, TE/bleeding, wound, and IAI). BEV, bevacizumab; GIP, gastrointestinal perforation; TE, thromboembolism.

**Table 1 jcm-12-02065-t001:** Clinico-pathologic features of enrolled patients.

			Overall (N = 79)
Age, year (range)			56.1 ± 12.0 (19–85)
BMI, kg/m^2^ (range)			23.1 ± 4.2 (16.9–38.4)
Charlson Comorbidity Index ^+^			
	0		47 (59.5)
	>0		32 (40.5)
		Mean score	1.6
		DM	9 (11.4)
		Hypertension	24 (30.4)
		Renal disease	2 (2.5)
		Peptic ulcer	3 (3.8)
		Others	13 (16.5)
Cancer type			
	Ovarian cancer		72 (91.1)
	Fallopian tube cancer		4 (5.1)
	Peritoneal cancer		3 (3.8)
FIGO Stage			
	I		6 (7.6)
	II		6 (7.6)
	III		47 (59.5)
	IV		20 (25.3)
Histology			
	High-grade serous		45 (59.2)
	Low-grade serous		3 (3.9)
	Clear cell		16 (21.1)
	Endometrioid		4 (5.3)
	Mucinous		2 (2.6)
	Mixed type or adenocarcinoma		6 (7.9)
	Unknown		3 (3.8)
Type of primary treatment			
	Upfront NACT + IDS ^#^		30 (38.0)
	Upfront surgery		45 (57.0)
	Unknown *		4 (5.1)
Type of surgery			
	Optimal debulking (residual tumor < 1 cm)		51 (64.6)
	Suboptimal debulking		24 (30.4)
	No surgery		4 (5.1)
Regimen of adjuvant chemotherapy			
	Platinum + Paclitaxel		57 (72.2)
	Platinum + Cyclophosphamide		4 (5.1)
	Others		7 (8.9)
	Not done		11 (13.9)
Response to platinum			
	Very sensitive (≥12 months)		23 (34.3)
	Sensitive (6–12 months)		21 (31.3)
	Resistant (<6 months)		22 (32.8)
	Unknown		1 (1.5)

^+^ The maximal score of Charlson Comorbidity Index: 37. ^#^ With or without using bevacizumab. * Patient referred from other institute for salvage chemotherapy but her previous disease information not clearly described. BMI, body mass index; DM, diabetes mellitus; FIGO, International Federation of Gynecology and Obstetrics; NACT, neoadjuvant chemotherapy; IDS, interval debulking surgery.

**Table 2 jcm-12-02065-t002:** Details of patients treated with bevacizumab.

			Overall (N = 79)
Role of Bevacizumab ^+^			
	Neoadjuvant		7 (8.9)
	Frontline		42 (53.2)
	Salvage		48 (60.8)
		Platinum sensitive	32
		Platinum resistant	14
		Unknown	2
Regimen combined with bevacizumab *			
	Combined regimen of chemotherapy		55 (69.6)
		Carboplatin + Paclitaxel	40
		Carboplatin + Lipodox	10
		Platinum + Gemcitabine	2
		Carboplatin + Topotecan	2
		Gemcitabine + Lipodox	1
	Single agent of chemotherapy		33 (41.8)
		Paclitaxel	6
		Lipodox	16
		Topotecan	8
		Cyclophosphamide	3
	Bevacizumab alone		2 (2.5)
Dose of Bevacizumab, mg/kg ^#^			
	<7.5		6 (7.6)
	7.5		64 (81.0)
	>7.5		10 (12.7)
Total dose of Bevacizumab, mg			4058 ± 3558
	Minimum		200
	Maximum		16838

^+,^ *^, #^ Avastin could be used in more than one clinical scenario.

**Table 3 jcm-12-02065-t003:** Assessing the adverse events after exposing to bevacizumab.

			Overall (N = 79)
Change in baseline blood pressure, n (%)			20 (25.3)
	negative history of hypertension		17 (21.5)
		Grade 1–2	14 (17.7)
		Grade ≥ 3	3 (3.8)
	history of hypertension		3 (3.8)
		Grade 1–2	2 (2.5)
		Grade ≥ 3	1 (1.3)
“De novo” proteinuria, n (%)			12 (15.2)
		Grade 1–2	12 (15.2)
		Grade ≥ 3	0
Gastrointestinal perforation, n (%)			4 (5.1)
		Grade 1–2	3 (3.8)
		Grade ≥ 3	1 (1.3)
Thromboembolic events/hemorrhage, n (%)			5 (6.3)
		Grade 1–2	3 (3.8)
		Grade ≥ 3	2 (2.5)
Wound-healing complications/fistula, n (%)			1 (1.3)
		Grade 1–2	1 (1.3)
		Grade ≥ 3	0 (0)
Intra-abdominal infection with unspecified etiology, n (%)			4 (5.1)
		Grade 1–2	4 (5.1)
		Grade ≥ 3	0

**Table 4 jcm-12-02065-t004:** Patients having adverse events after exposure to bevacizumab (BEV).

A. Patients having gastrointestinal perforation (GIP) after exposure to BEV													
Case	Age	Stage Histology	Role of BEV	Surgery	History of peptic ulcer or IBS	Symptoms of bowel involvement	Bowel resection (at debulking surgery)	Bowel involvement on image	BEV usage		Site of GIP	Management for GIP	Survival days after GIP
									Dose (mg/kg) (triweekly)	Cycle			
1	49	IVB HGSC	Salvage	Optimal (no residual)	+	+	-	+	7.5	15	No definite site	Exploratory laparotomy * followed by Conservative tx ^#^	116
2	51	IVB HGSC	Frontline	Suboptimal	-	+	-	+	10	5	No definite site	Conservative tx ^#^	6
3	61	IVA HGSC	Neoadjuvant	Optimal (<1cm)	-	+	-	+	7.5	1	Favor small intestine	Conservative tx ^#^	603
4	52	IIB CCC	Salvage	Optimal (<1cm)	+	-	+	+	7.5	4	No definite site	Conservative tx ^#^	9
**B. Patients having thromboembolism (TE) after the exposure to BEV**													
Case	Age	Stage Histology	BMI	Charlson Comorbidity Index	Medical history	Role of BEV	BEV usage		Site of TE		Management	Survival days after TE	
							Dose (mg/kg)(triweekly)	Cycle					
1 ^†^	52	IIB CCC	21.3	1	HTN	Salvage	7.5	4	Right distal ICA		Thrombectomy	12	
2	53	IIIC HGSC	22	0	Nil	Salvage	7.5	15	Left frontal lobe		Conservative	29	
**C. Patient having bleeding episodes after the exposure to BEV**													
Stage Histology		BMI	Charlson Comorbidity score		Medical Hx	Role o BEV	BEV dose	Site of bleeding			Management		
IVB CCC		19.1	2		Gastric ulcer Cirrhosis		7.5 mg/kg triweekly	Left renal subcapsular hematoma			Hold BEV temporarily		
IIIC HGSC		23.3	1		HTN		7.5 mg/kg triweekly	Gum and nasal mucosa bleeding			Hold BEV temporarily		
IIIC HGSC		21.1	0		Nil		7.5 mg/kg triweekly	Metastatic liver tumor bleeding			Tranexamic acid, blood transfusion, TAE		
**D. Patient having surgical wound-healing complications or fistula after the exposure to BEV**													
Case	Age	Stage Histology	BMI	Medical Hx	Surgery	Role of BEV	BEV dose	Interval between last BEV and surgery (days)			Wound complication	Management	
1	62	IIIC HGSC	22.9	Nil	Optimal debulking		7.5 mg/kg triweekly	50			Wound infection with intra-abdominal abscess	CT-guided drainage Antibiotics	
**E. Patient having intra-abdominal infection after the exposure to BEV**													
Case	Age	Stage Histology		BMI	Charlson Comorbidity score	Medical Hx	Surgery	Role of BEV			BEV dose	Management	
1	66	IVB LGSC		19.2	0	Nil	Nil	Frontline, Salvage			7.5 mg/kg triweekly	Conservative tx ^#^	
2	60	IIIC LGSC		23.7	2	HTN, DM	Optimal debulking	Salvage			5 mg/kg triweekly	Conservative tx ^#^	
3	83	IIIB Adenocarcinoma		22.6	2	Gastric ulcer, Cirrhosis	Nil	Neoadjuvant			7.5 mg/kg triweekly	Conservative tx ^#^	
4	66	IIIC HGSC		23.0	3	DM, Lymphoma	Optimal debulking	Frontline			10 mg/kg triweekly	Conservative tx ^#^	

* Exploratory laparotomy: drainage of intra-abdominal abscess and adhesiolysis. ^#^ Conservative tx, conservative treatment includes Nulla per os (NPO), total parenteral nutrition (TPO), and broad-spectrum antibiotics. ^†^ Case 1 of Table 4B also refer to case 4 of Table 4A. BEV, bevacizumab; GIP, gastrointestinal perforation; HGSC, high-grade serous carcinoma; LGSC, low-grade serous carcinoma; CCC, clear cell carcinoma; IBS, irritable bowel syndrome; NACT, neoadjuvant chemotherapy; TE, thromboembolism; BMI, body mass index; ICA, internal carotid artery; NPO, nulla per os; TPN, total parenteral nutrition; TAE, trans-arterial embolization; CT, computer tomography; HTN, hypertension; DM, diabetes mellitus.

## Data Availability

The data for supporting the findings in this study are available from the corresponding author upon reasonable request.

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
