# Peer review of "Safety Analysis of Bevacizumab in Ovarian Cancer Patients"

_jcm, 2023, doi:10.3390/jcm12052065_

Round 1

Reviewer 1 Report

Please separate stage I and II Figo

Your are mentioning several times, risks factors for developing GIP, but I do not find those risks factors in your manuscript.

You have an important proportion of suboptimal cytoreduction meaning that you have bad surgeons or more likely that your population have worse diseases than those published in other trials. How did you decide to add bevacizumab, does all your patients received bavacizumab if no contra indication?

Reviewer 2 Report

This is a retrospective analysis of side effects of bevacizumab in 79 patients from Taiwan. The main reason to perform such an analysis is the fact that most studies contains only few Asian women and effect and side effects of drugs can differ between Western patients and Asian patients.

The study includes both women receiving bevacizumab (bev) in first line and women treated in the relapse setting. In all cases, bev was given together with chemotherapy.

Bev was given in first line and again at relapse in 11 patients.

The frequencies of side effects found in this study are in line with frequencies found in Western RCT on bev and so is the type of side effects.

Comments:

The study population is small giving wide confidence intervals for the findings.

11 patients received bev on two occasions (presumably both during first line treatment and at relapse. This should be more clearly mentioned and discussed with reference to literature on this topic.

Line 39-40:  “Patients with recurrent EOC are usually further classified by the platinum-free interval (PFI)” is mentioned twice.

Line 121/122: “When analyzing the response to platinum, 65.6% of the patients were platinum-sensitive, and 32.8% 122 were platinum-resistant”. At this point the authors probably refer to relapses after first line treatment being either platinum sensitive or platinum resistant. The text should be reviewed to describe the authors intention.

Line 136-137; “81.0% of the BEV dose 136 was prescribed at 7.5 mg/kg”. It should be added that this dose was given with 3 weeks interval.

Section 3.4:

The use of ROC curves to illustrate the correlation between hypertension and proteinuria does not seem appropriate in such a small study.

The discussion on bev related hypertension and proteinuria seems relevant (section 4.1 and 4.2). The RCT Boost study (J. Phisterer, DOI https://doi. org/10.1200/JCO.22. 0101) should be included in the discussion as it deals with prolonged use of bev.

It is notably that all 4 patients with gastrointestinal perforations in fact had symptoms of bowel involvement and thereby contraindication to treatment with bev. A warning against the use of bev to such patients would be appropriate.
